# A Novel Non-Invasive Biomarker for Gastric Cancer: Monocyte-to-HDL Ratio and Clinicopathological Parameters in Predicting Survival Outcomes

**DOI:** 10.3390/cancers17172816

**Published:** 2025-08-28

**Authors:** Mehmet Salim Demir, Gözde Ağdaş

**Affiliations:** Department of Medical Oncology, Van Training and Research Hospital, 65300 Van, Turkey; drgozdeagdas@gmail.com

**Keywords:** gastric cancer, monocyte-to-HDL ratio, prognosis, survival outcomes, biomarker, inflammation

## Abstract

Gastric cancer is a significant cause of cancer-related deaths worldwide. This study investigated the preoperative monocyte-to-high-density lipoprotein cholesterol ratio (MHR) as a simple and low-cost prognostic marker in patients undergoing curative gastrectomy. We found that patients with higher MHR values had worse overall and progression-free survival. Elevated MHR remained an independent predictor of disease progression even after adjusting for other factors. Since MHR can be calculated from routine blood tests, it may serve as an accessible tool to help identify high-risk patients and guide personalized treatment strategies.

## 1. Introduction

Gastric cancer (GC) remains a global health burden, ranking among the leading causes of cancer-related mortality worldwide [1]. Despite advancements in surgical techniques and systemic therapies, the prognosis for advanced GC remains poor, highlighting the need for reliable prognostic biomarkers to guide treatment strategies [2,3]. Systemic inflammation plays a crucial role in cancer progression, and various inflammation-based indices have been investigated for their predictive value in GC [3,4,5]. The monocyte-to-high-density lipoprotein cholesterol ratio (MHR) has emerged as a novel marker reflecting the balance between pro-inflammatory monocytes and anti-inflammatory HDL [6,7].

Elevated MHR levels have been associated with worse outcomes in several malignancies, suggesting their potential as a prognostic indicator in GC [7,8]. Recent studies have investigated the prognostic significance of MHR in patients with GC undergoing curative gastrectomy [9]. Liu et al. reported that elevated MHR levels correlated with advanced tumor stages and unfavorable clinicopathological features [10]. These findings underscore the potential utility of MHR as a cost-effective and straightforward biomarker for risk stratification in GC. In addition to MHR, other systemic inflammation markers such as the neutrophil-to-lymphocyte ratio, platelet-to-lymphocyte ratio, and systemic immune-inflammation index have been studied for their prognostic relevance in GC [11,12,13]. These markers, derived from routine blood tests, offer valuable insights into the host’s inflammatory response and tumor biology [14]. The integration of MHR and other inflammation-based indices into practice requires further validation [9,15,16].

The present study aimed to evaluate the impact of MHR, lymphovascular invasion, perineural invasion, signet ring cell histology, and TNM staging on survival outcomes in patients undergoing surgery for gastric cancer. By identifying these prognostic factors, the study aimed to enhance the accuracy of survival prediction and contribute to the development of personalized treatment strategies.

## 2. Materials and Methods

### 2.1. Study Design and Patient Population

This retrospective cohort study was conducted at the Departments of Surgical Oncology and Internal Medicine Oncology and adhered to the Strengthening the Reporting of Observational Studies in Epidemiology (STROBE) guidelines. The population consisted of 304 patients who underwent curative-intent gastrectomy for histologically confirmed gastric adenocarcinoma between January 2018 and February 2025. Ethical approval was obtained from the institutional review board (Decision No: GOKAEK/2025-02-17; Date: 28 February 2025), and the study complied with the Declaration of Helsinki. Written informed consent was obtained from all patients before surgery, and study data were de-identified using unique study codes, with the re-identification key stored on a separate, encrypted server accessible only to the principal investigator (PI).

### 2.2. Inclusion and Exclusion Criteria

Patients were eligible for inclusion if they were 18 years or older and had a confirmed diagnosis of gastric adenocarcinoma, as determined by histopathological examination. All patients underwent D1+ or D2 lymphadenectomy, as determined by tumor location, stage, and surgeon’s discretion, following the guidelines of the Japanese Gastric Cancer Association. They had no evidence of distant metastasis (M0) based on preoperative imaging, including contrast-enhanced computed tomography (CT) scans and positron emission tomography (PET) scans. Additionally, only patients with complete preoperative blood test results, including a full blood count and lipid panel obtained within 7 days before surgery, were considered for analysis. Patients were excluded if they had any history of another active malignancy or a prior cancer diagnosis within the past five years, or if they had evidence of acute infection, systemic inflammation, autoimmune disorders, chronic liver disease, or were receiving immunosuppressive therapy at the time of surgery. Cases with incomplete data, positive surgical margins (R1 or R2 resection), or those who died within 30 days postoperatively were also excluded from the final cohort to eliminate the influence of perioperative mortality. Additionally, baseline comorbidities were retrospectively collected from preoperative anesthesia and surgical assessments. The Comorbidity Index was calculated for each patient. Specific comorbidities, including cardiovascular disease, diabetes mellitus, and chronic kidney disease, were recorded and analyzed as potential confounders in survival models.

### 2.3. Preoperative Assessment and Laboratory Parameters

Preoperative evaluations included a comprehensive physical examination, laboratory tests, chest and abdominopelvic CT scans, endoscopy with biopsy, and optional PET-CT scans for staging. Peripheral blood samples were collected after an overnight fast (8–10 h) within one week before surgery. Blood samples were analyzed using a Sysmex XN-1000 hematology analyzer (Sysmex Corporation, Kobe, Japan) for complete blood count (CBC), and HDL cholesterol was measured using the Beckman Coulter AU5800 chemistry analyzer (Beckman Coulter, Brea, CA, USA). The MHR was calculated by dividing the absolute monocyte count by the HDL cholesterol (mg/dL).

### 2.4. Surgical Procedure and Pathological Evaluation

Pathologists were blinded to clinical data to prevent potential observer bias in evaluating semi-quantitative features such as lymphovascular and perineural invasion, ensuring that histopathological assessment was based solely on morphological criteria. Surgical treatment consisted of total or subtotal gastrectomy based on tumor location and extent, with D_1_+ or D_2_ lymph node dissection following Japanese Gastric Cancer Association guidelines. All resected specimens were examined by experienced gastrointestinal pathologists blinded to the clinical data. Histologic subtypes were classified based on the Lauren classification and the presence or absence of signet-ring cell carcinoma (SRC). Pathological TNM staging was performed according to the 8th edition of the American Joint Committee on Cancer classification. Lymphovascular invasion was defined as the presence of tumor emboli within endothelial-lined vascular or lymphatic channels. Perineural invasion was recorded if tumor cells were seen within any of the three layers of the nerve sheath or close to nerve fibers.

### 2.5. Follow-Up Protocol

Postoperative surveillance was standardized across all patients. Follow-up visits were scheduled every 3 months for the first 2 years, every 6 months for years 3 to 5, and annually thereafter. Routine evaluations included a physical examination, complete blood count, liver function tests, carcinoembryonic antigen, carbohydrate antigen 19-9, and a contrast-enhanced thoracoabdominal CT scan every 6 months. Disease progression was defined as locoregional recurrence or distant metastasis, as confirmed by imaging or histological evidence. Overall survival (OS) was defined as the time from surgery to death from any cause. Progression-free survival (PFS) was defined as the time from surgery to tumor recurrence, metastasis, or death, whichever occurred first.

### 2.6. Statistical Analysis

All statistical analyses were conducted using IBM SPSS Statistics for Windows, Version 25.0 (IBM Corp., Armonk, NY, USA). Continuous variables were tested for normality using the Shapiro–Wilk and Kolmogorov–Smirnov tests. Variables with normal distribution are expressed as mean ± standard deviation (SD), while non-normally distributed variables are presented as median. Categorical variables are reported as frequencies and percentages. The optimal cutoff value for MHR to predict survival was determined using receiver operating characteristic (ROC) curve analysis. The calculated cutoff of 11.02 reflects both the unit conventions applied in our dataset and the optimization method (Youden index). Differences influence variability in cutoff values across various cancer types, analytical platforms, patient populations, and statistical derivation methods. External validation and sensitivity analyses are required to confirm the robustness of this threshold. Patients were subsequently dichotomized into high- and low-MHR groups based on this threshold. Survival curves were generated using the Kaplan–Meier method and compared with the log-rank test. Cox proportional hazards regression analyses were used to identify independent predictors of OS and PFS. The number of OS events per variable in the multivariate Cox model was 12.3, satisfying the >10 EPV rule. Variance inflation factors were all <2.0, excluding problematic multicollinearity. Backward elimination yielded the same outcome, with no independent predictors for OS. Variables with a univariate *p*-value < 0.05 were included in the multivariate Cox models. Schoenfeld residuals verified proportional hazard assumptions. Multicollinearity was assessed using variance inflation factors (VIFs); a VIF > 5.0 was considered indicative of multicollinearity, and all VIF values in our models were <2.0. Missing data were handled using multiple imputation under a wholly conditional specification model, assuming data were missing at random (MAR). Hazard ratios (HRs) and 95% confidence intervals (CIs) were reported. To address potential overfitting and assess model stability, internal validation was performed using bootstrap resampling (1000 iterations), which demonstrated minimal optimism in hazard ratio estimates, indicating good model reproducibility within the study cohort. All statistical tests were two-tailed, and a *p*-value < 0.05 was considered significant.

### 2.7. Ethical Considerations

This study was approved by the Non-Interventional Clinical Research Ethics Committee of Van Training and Research Hospital (Approval ID: GOKAEK/2025-02-17). Informed consent was obtained from all participants for the use of their data in anonymized research. Data were collected following national data protection regulations and securely stored on encrypted institutional servers, accessible only to the study investigators. All analyses were performed on de-identified datasets in accordance with institutional policy and national data-protection regulations; no direct identifiers were retained.

## 3. Results

A comprehensive analysis of sociodemographic and clinicopathological characteristics among patients undergoing surgery for gastric cancer revealed that the majority were over the age of 50 (94.4%), with a predominance of male patients (64.5%). The mean age was 62.8 ± 8.4 years (range: 38–82 years). Regarding functional status, more than half of the patients (53.9%) had an Eastern Cooperative Oncology Group performance score of 0, indicating a generally favorable physical condition. Most patients had a normal body mass index (69.4%). Histopathological evaluation identified the intestinal subtype of the Lauren classification as the most common (71.2%), while signet ring cell histology was observed in approximately one-quarter of cases (25.4%). According to staging (Table 1), the majority of patients were categorized as stage III (58.9%), which was also consistent with the pathological staging results (62.2%). A total of 61 patients (20.1%) received both neoadjuvant and adjuvant chemotherapy.

In 75.3% of patients, the number of lymph nodes retrieved during surgery exceeded 15. LVI and PNI were detected in 62.8% and 61.8% of cases, respectively. The median follow-up duration was calculated as 56 months. The prognostic value of the MHR in predicting mortality was assessed using the ROC curve (Figure 1). The area under the curve was 0.654 (95% CI: 0.59–0.718), and the optimal cut-off was determined to be ≥11.02. At this threshold, sensitivity and specificity were 62.6% and 62.4%, respectively. Of the 304 patients, 152 (50.0%) had MHR ≥ 11.02 and 152 (50.0%) had MHR < 11.02.

Patients with a monocyte/HDL ratio ≥ 11.02 had a 5-year OS rate of 51.4%, compared to 72.2% in those with a ratio < 11.02. The overall 5-year OS in the cohort was 61.5%. In patients with signet ring cell carcinoma, the 5-year OS rate was 47.5%, lower than the 65.4% observed in those without this histological feature. Five-year OS rates according to clinical TNM stage were as follows: stage I: 83.5%, II: 69.4%, III: 53.6%, and IV: 33.3%. In the presence of LVI and PNI, the 5-year OS rates were 50.6% and 53.3%, respectively. In their absence, the rates increased to 80.4% and 75%, respectively (Table 2). Among the 123 deaths recorded, 104 were attributed to gastric cancer progression or metastasis, while 19 were due to non-cancer causes, predominantly cardiovascular events and severe infections. Age was incorporated into multivariate survival models; it was not independently associated with PFS or OS after adjustment for stage and MHR. Higher ECOG scores were associated with worse OS and PFS in univariate analysis, but did not retain independent prognostic significance after adjusting for stage and MHR. Treatment heterogeneity represents a potential confounder in the MHR–survival association. As MHR is measured preoperatively, subsequent perioperative treatments—particularly neoadjuvant and adjuvant chemotherapy—may modify recurrence risk and overall prognosis, potentially attenuating the biomarker’s impact on OS.

Progression-free survival analyses identified signet ring cell histology, stage, neoadjuvant chemotherapy, pathological stage, lymphovascular invasion (LVI), perineural invasion (PNI), adjuvant chemotherapy, and metastasis-free rate (MFR) as prognostic factors. The 5-year PFS rate was 55.6% in patients with signet ring cells and 76.2% in those without this feature. According to pathological staging, the 5-year PFS rates were 94.4% for stage I, 85.2% for stage II, and 63.6% for stage III. The presence of LVI and PNI was associated with reduced 5-year PFS (65.6% vs. 86.4% and 64% vs. 89.2%, respectively). Similarly, patients with MHR ≥ 11.02 had a 5-year PFS rate of 65.2%, compared to 80.5% in those with lower ratios (Table 3).

As shown in Figure 2, individuals who received neoadjuvant chemotherapy (red line) demonstrated significantly lower PFS rates than those who did not (blue line) (67.1% vs. 76.7%, *p* = 0.038) (Figure 2a). Likewise, patients treated with adjuvant chemotherapy (red line) had worse PFS than those without adjuvant treatment (blue line) (68.6% vs. 82.3%, *p* = 0.014) (Figure 2b). In multivariate Cox regression analysis for overall mortality, the variables identified as significant in univariate analysis—signet ring cell histology, clinical TNM stage, pathological stage, number of dissected lymph nodes, LVI, PNI, and MHR—were included. The median Comorbidity Index was 3 (IQR 2–4). Cardiovascular disease was present in 21.7% of patients, diabetes mellitus in 18.1%, and chronic kidney disease in 4.6%. In multivariate Cox models, neither the Comorbidity Index nor any individual comorbidity showed a significant association with PFS or OS, and the inclusion of these parameters did not alter the hazard ratio for MHR (Table 4). Conversely, in the multivariate analysis for progression, only MHR ≥ 11.02 emerged as an independent predictor. Elevated MHR had a 1.93-fold increased risk of progression (HR: 1.93, 95% CI: 1.17–3.18, *p* = 0.010) (Table 5).

## 4. Discussion

The present study provides single-center evidence suggesting that the preoperative MHR is a statistically significant but moderately accurate predictor of disease progression in patients undergoing curative-intent gastrectomy for gastric adenocarcinoma. Given the moderate discrimination observed (AUC = 0.654; 95% CI 0.59–0.718), the clinical utility of MHR is best considered adjunctive to established clinicopathologic factors rather than standalone. While MHR retained independent prognostic value for progression-free survival, its moderate discriminative capacity suggests that its clinical utility is likely maximized when used in conjunction with established clinicopathological parameters rather than as a standalone biomarker. Our analysis, based on a cohort of all patients who underwent D1+ or D2 lymphadenectomy according to tumor location, stage, and surgeon discretion, consistent with Japanese Gastric Cancer Association guidelines, identified higher preoperative MHR as being associated with a nearly twofold increased risk of progression. While MHR demonstrated a strong association with PFS, its significance for OS was attenuated in the multivariate model. Similarly, the well-established adverse prognostic indicators of SRC histology, LVI, and PNI, all of which were highly significant in univariate analyses, did not retain independent predictive power for OS after adjustment for clinicopathological confounders, most notably pathological TNM.

The biological plausibility of MHR as a prognostic signal is supported—though not proved—by the context-dependent roles of circulating monocytes/macrophages and HDL in tumor biology and systemic inflammation. Monocytes can differentiate into tumor-associated macrophages that promote angiogenesis and immune evasion, whereas HDL participates in multiple metabolic and immunomodulatory pathways; these functions are not strictly antagonistic and vary depending on the disease context [17]. In gastric cancer, tumor-associated macrophages typically exhibit a pro-angiogenic phenotype, promoting tumor cell proliferation, invasion, and metastasis, while supporting neovascularization through the secretion of factors such as VEGF-A, thereby facilitating tumor growth and dissemination [18]. Conversely, high levels of HDL-C have been inversely associated with gastric cancer-related mortality; in an extensive cohort study, increasing HDL-C concentrations were correlated with reduced mortality, and this protective effect was also observed in subgroups undergoing gastrectomy or endoscopic resection [19]. These findings substantiate the biological mechanisms underpinning the prognostic role of MHR and reinforce its conceptualization as a biomarker that reflects the dynamic balance between monocyte-driven tumor progression and the protective effects of HDL. A recent meta-analysis of randomized controlled trials by Granieri et al. demonstrated that cytoreductive surgery combined with hyperthermic intraperitoneal chemotherapy confers a significant survival benefit in patients with locally advanced gastric cancer [20]. It underscores the importance of multimodal prognostic factors when stratifying risk and tailoring postoperative management. MHR functions as an elegant and simple composite biomarker that captures the systemic balance between the pro-tumorigenic inflammatory drive, represented by monocytes, and the host’s protective anti-inflammatory and metabolic state, reflected by HDL [21]. Although preoperative MHR demonstrated significance as a prognostic marker, its AUC of 0.654 denotes only moderate discriminative capacity. This consideration supports using MHR as an adjunct rather than a stand-alone marker. Future prognostic models combining MHR with established clinicopathological variables may enhance predictive performance and facilitate actionable stratification.

Our study identified a preoperative MHR ≥ 11.02 as an independent predictor of disease progression, conferring a hazard ratio of 1.93. This finding aligns with a growing body of evidence supporting the prognostic role of MHR in various types of cancer. However, direct comparison across studies is complicated by significant heterogeneity in reported cutoff values, which represents a major challenge for its clinical implementation. Our calculated cutoff of 11.02, derived using monocyte counts in cells/µL and HDL cholesterol in mg/dL via Youden index optimization, is a statistically robust threshold for our cohort. The literature, however, presents a broad spectrum of cutoffs, such as 1.184 in pancreatic cancer and 0.3 in non-small cell lung cancer [22]. This apparent disparity is likely attributable to several factors. Differences in the statistical methods used to determine the cutoff and inherent biological differences between cancer types and patient populations may contribute to this heterogeneity [23]. This lack of a standardized MHR threshold is a significant barrier to its widespread clinical adoption, underscoring the urgent need for consensus guidelines on its calculation and interpretation —a challenge that also affects other inflammation-based ratios, such as the lymphocyte-to-monocyte ratio [24]. A central and intriguing finding of our study is the dissociation between MHR’s predictive capacity for PFS and its lack of independent significance for OS. We observed that preoperative MHR predicted PFS but did not retain independent significance for OS after multivariable adjustment. This divergence is not unexpected in gastric cancer, where post-recurrence treatments and survivorship heterogeneity can attenuate the relationship between baseline risk and all-cause mortality. Consequently, we report this as a descriptive finding consistent with contemporary therapeutic practice, rather than as a novel mechanistic claim.

While an elevated MHR strongly heralded earlier disease recurrence, it was not an independent predictor of overall mortality in our multivariate model. This observation may not indicate a failure of the biomarker but rather reflects the evolving landscape of gastric cancer management and the complex nature of OS as an endpoint. It is essential to note that although MHR ≥ 11.02 was independently associated with an increased risk of progression, the moderate AUC and balanced sensitivity/specificity suggest that its interpretation should be done cautiously. Although some between-group PFS differences reached statistical significance, the absolute differences were modest and potentially confounded by non-random treatment allocation. Accordingly, their clinical significance should be interpreted cautiously. In practical terms, MHR may help refine risk stratification but is unlikely to supplant existing prognostic indices or staging systems. Its role is best envisioned as part of a composite predictive model, where it can contribute prognostic information.

In the current era, characterized by the availability of effective multi-line systemic therapies, the interval between disease progression and death can be significantly prolonged [25]. The advent of salvage chemotherapy, targeted agents, and immune checkpoint inhibitors for metastatic gastric cancer means that disease recurrence is no longer an immediate harbinger of mortality for many patients [26,27]. An effective salvage regimen administered post-progression could mitigate the initial poor prognosis conferred by a high preoperative MHR, thereby uncoupling the statistical association between the baseline biomarker and the final OS endpoint. MHR ≥ 11.02 was associated with increased risk of progression (adjusted HR = 1.93; 95% CI 1.17–3.18), yet its discrimination was moderate; therefore, MHR should be viewed as a supportive preoperative risk marker rather than a definitive indicator of treatment failure. This contrasts with findings in other malignancies, such as NSCLC, where MHR was a powerful independent predictor of OS, perhaps reflecting differences in disease biology or the relative efficacy of salvage treatments in that setting [10]. Clinically, a high preoperative MHR could be integrated into a multimodal prognostic model to stratify patients for tailored follow-up and adjuvant treatment strategies. For example, patients with elevated MHR might be prioritized for early postoperative surveillance imaging, expedited initiation of adjuvant therapy, or enrollment in clinical trials evaluating intensified perioperative regimens. We do not propose withholding surgery based solely on MHR, but instead use it as a supportive tool in individualized treatment.

The field of inflammatory biomarkers in oncology is extensive, featuring well-studied indices such as SII and SIRI [11]. Recent meta-analyses have solidified the prognostic value of these markers in gastric cancer. Yang et al., encompassing 30 studies, reported a pooled hazard ratio of 1.53 for high SII and poor OS, and a pooled HR of 1.41 for poor recurrence-free survival [14]. Notably, the hazard ratio of 1.93 for PFS observed in our high-MHR is higher than the pooled RFS estimate for SII. Our findings add to the evidence that inflammation-based indices—including MHR—carry prognostic information in gastric cancer; formal head-to-head comparisons and externally validated models are needed to contextualize relative predictive performance. Given that MHR is calculated from a simple lipid panel and complete blood count—tests that are routinely performed and universally available—it represents a highly accessible, cost-effective, and potentially more potent tool for identifying patients at high risk of recurrence. In our univariate analysis, SRC histology was associated with worse OS, with a 5-year rate of 47.5% compared to 65.4% for non-SRC tumors. This is consistent with its clinical reputation as an aggressive subtype. However, SRC failed to maintain significance in the multivariate model.

The prognostic significance of SRC is one of the most debated topics in gastric oncology, with many studies reporting a stage-dependent effect: a paradoxical or similar prognosis in early-stage disease but a worse prognosis in advanced stages [28]. Our multivariate Cox regression, which adjusts for the influence of these covariates, likely demonstrated that the pathological TNM stage itself more accurately and powerfully captured the prognostic information carried by the “SRC” label. Similarly, both LVI and PNI were potent predictors of poor survival in our univariate analyses. This finding highlights the role of LVI and PNI as mechanistic pathways, rather than ultimate prognostic endpoints, in a comprehensive statistical model. An emerging area of research suggests that while LVI or PNI alone may have their prognostic value attenuated by TNM stage, their concomitant presence may represent a phenotype of exceptional biological aggressiveness that retains independent significance. A study by Hwang et al. found that LVI+/PNI+ status was a powerful independent predictor of poor OS in stage II/III GC patients who received adjuvant chemotherapy [29]. Similarly, Blumenthaler et al. found that concurrent LVI/PNI was associated with decreased survival in patients treated with preoperative therapy [30]. Our study was not designed to test this specific interaction, as LVI and PNI were entered as separate variables.

This study’s strengths include its well-characterized, homogenous patient cohort, where all individuals underwent standardized surgical procedures and were managed within a single high-volume system, which minimizes treatment heterogeneity. The long-term median follow-up of 56 months provides a robust dataset for survival analysis. Nevertheless, several limitations must be acknowledged. The retrospective nature of our study inevitably introduces the potential for selection bias, particularly in patient inclusion and data completeness. Although we attempted to mitigate this risk by applying strict inclusion/exclusion criteria, conducting the study within a single high-volume center, and implementing standardized surgical and follow-up protocols, residual confounding cannot be entirely excluded. Future prospective multicenter studies are warranted to validate these findings in a broader population. Although the MHR demonstrated statistical significance as a prognostic marker, its discriminative capacity (AUC: 0.654) indicates only moderate accuracy. This finding underscores the importance of integrating MHR into multi-parameter prognostic models, rather than relying solely on it as a standalone predictor. Such combined approaches may improve predictive performance and clinical applicability. The potential confounding effect of comorbidities, particularly cardiovascular disease, was addressed by including the Comorbidity Index and individual comorbidities in multivariate analyses. These adjustments did not significantly alter the prognostic effect of MHR, suggesting that its predictive capacity is not solely attributable to the comorbidity burden. The sensitivity and specificity values (62%) underscore that MHR should not be applied in isolation for prognostication. Its optimal utility may be in combination with established clinical and pathological variables within composite prediction models, where even moderate individual predictors can improve model calibration and risk stratification. Ultimately, further basic and translational research is necessary to elucidate the precise biological mechanisms underlying the relationship between a high MHR and tumor progression.

## 5. Conclusions

In conclusion, although the MHR demonstrated significance as a prognostic marker in our cohort, its discriminative ability was moderate. Consequently, MHR should not be regarded as a definitive predictor in isolation but rather as a cost-effective, readily obtainable adjunct within a broader preoperative risk assessment framework. Integration with other inflammation-based and clinicopathological factors may enhance predictive performance and clinical applicability. Future prospective multicenter studies are warranted to validate these findings and to establish standardized cutoff values and analytic methods for MHR before it can be confidently integrated into routine decision-making.

## Figures and Tables

**Figure 1 cancers-17-02816-f001:**
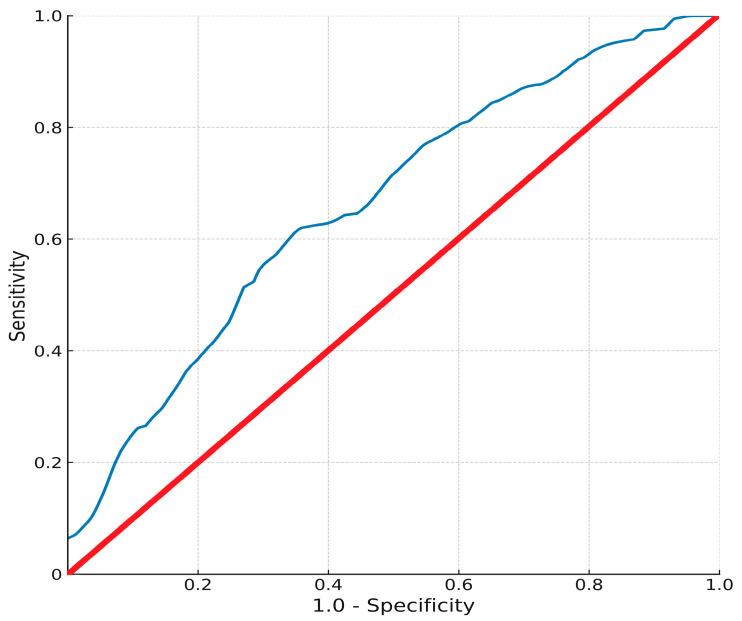
Receiver operating characteristic curve for preoperative MHR predicting 5-year overall survival. Blue curve: sensitivity vs. 1–specificity for MHR; red diagonal: no-discrimination line (AUC = 0.654; 95% CI 0.59–0.718; sensitivity 62.6%; specificity 62.4% at cut-off of 11.02).

**Figure 2 cancers-17-02816-f002:**
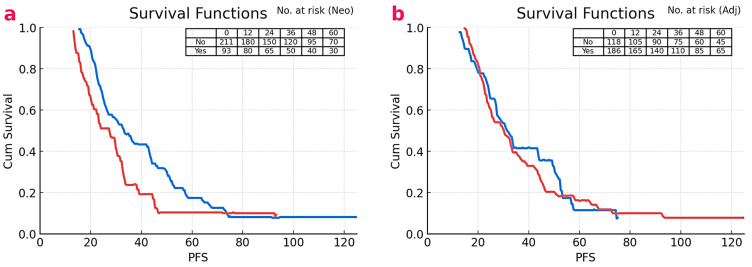
Kaplan–Meier progression-free survival (PFS). (**a**) Neoadjuvant chemotherapy status (red = received; blue = no neoadjuvant)—log-rank *p* = 0.038. (**b**) Adjuvant chemotherapy status (red = received; blue = no adjuvant)—log-rank *p* = 0.014. Numbers at risk for each time point are shown in each panel. Note: Treatment groups are non-random, and between-group differences should be interpreted with caution due to potential confounding factors.

**Table 1 cancers-17-02816-t001:** Demographic and Clinicopathological Characteristics.

Variables	n: 304	%
Lauren classification		
▪ *Diffuse*	52	23.7
▪ *Intestinal*	156	71.2
▪ *Mixed*	7	3.2
▪ *Not available*	4	1.8
Signet-ring cell histology		
▪ *Absent*	197	74.6
▪ *Present*	67	25.4
Clinical TNM stage		
▪ *I*	38	12.5
▪ *II*	81	26.6
▪ *III*	179	58.9
▪ *IV*	6	2.0
Pathological stage		
▪ *I*	58	19.1
▪ *II*	57	18.8
▪ *III*	189	62.2
Retrieved lymph nodes		
▪ *<15*	75	24.7
▪ *≥15*	229	75.3
Lymphovascular invasion		
▪ *Absent*	113	37.2
▪ *Present*	191	62.8
Perineural invasion		
▪ *Absent*	116	38.2
▪ *Present*	188	61.8
Neoadjuvant chemotherapy		
▪ *No*	211	69.4
▪ *Yes*	93	30.6
Adjuvant chemotherapy		
▪ *No*	118	38.8
▪ *Yes*	186	61.2
Adjuvant radiotherapy		
▪ *No*	193	64.5
▪ *Yes*	106	35.5
Progression during follow-up		
▪ *No*	230	75.7
▪ *Yes*	74	24.3
Status at last follow-up		
▪ *Alive*	181	59.5
▪ *Deceased*	123	40.5

Values are presented as numbers (percent).

**Table 2 cancers-17-02816-t002:** Overall Survival Stratified by Key Clinicopathological Variables.

Variable	2-Year Survival (%)	5-Year Survival (%)	Median OS (Months)	Log-Rank *p*
Monocyte-to-HDL *< 11.02*	84.0	72.2	N-R	<0.001
Monocyte-to-HDL *≥ 11.02*	68.7	51.4	86.0	
Signet-ring cell *absent*	82.7	65.4	N-R	0.007
Signet-ring cell *present*	62.7	47.5	41.9	
Clinical *Stage I*	94.7	83.5	N-R	0.001
Clinical *Stage II*	83.9	69.4	N-R	
Clinical *Stage III*	70.8	53.6	137.7	
Clinical *Stage IV*	50.0	33.3	17.9	
Pathological *Stage I*	94.8	84.0	N-R	<0.001
Pathological *Stage II*	87.7	79.2	N-R	
Pathological *Stage III*	68.2	49.7	57.0	
Retrieved LNs *< 15*	85.3	73.5	N-R	0.028
Retrieved LNs *≥ 15*	74.6	57.4	174.3	
LVI *absent*	91.0	80.4	N-R	<0.001
LVI *present*	69.1	50.6	86.0	
PNI *absent*	85.3	75.0	N-R	<0.001
PNI *present*	71.8	53.3	103.0	

OS, overall survival; N-R: Not reached; CI, confidence interval; LNs, lymph nodes; LVI, lymphovascular invasion; PNI, perineural invasion. MHR ≥ 11.02, n = 152/304; MHR < 11.02, n = 152/304.

**Table 3 cancers-17-02816-t003:** Progression-Free Survival Stratified by Key Clinicopathological Variables.

Variable	2-Year PFS (%)	5-Year PFS (%)	Median PFS (Months)	Log-Rank *p*
Overall cohort	81.2	73.7	N-R	0.008
Signet-ring cell *absent*	83.5	76.2	N-R	
Signet-ring cell *present*	65.8	55.6	N-R	
*Clinical Stage I*	91.4	94.7	N-R	0.002
*Clinical Stage II*	82.4	82.4	N-R	
*Clinical Stage III*	75.2	65.3	N-R	
*Clinical Stage IV*	80.0	53.3	N-R	
Neoadj. chemotherapy—*No*	84.4	76.7	N-R	0.038
Neoadj. chemotherapy—*Yes*	73.7	67.1	N-R	
Pathological *Stage I*	94.4	94.4	N-R	<0.001
Pathological *Stage II*	94.5	85.2	N-R	
Pathological *Stage III*	72.7	63.6	N-R	
LVI *absent*	91.6	86.4	N-R	<0.001
LVI *present*	74.7	65.6	N-R	
PNI *absent*	90.6	89.2	N-R	<0.001
PNI *present*	75.4	64.0	N-R	
Adj. chemotherapy—*No*	87.6	82.3	N-R	0.014
Adj. chemotherapy—*Yes*	77.5	68.6	N-R	
Monocyte-to-HDL *< 11.02*	86.9	80.5	N-R	0.002
Monocyte-to-HDL *≥ 11.02*	74.5	65.2	N-R	

PFS, progression-free survival; N-R, Not reached; LVI, lymphovascular invasion; PNI, perineural invasion. MHR ≥ 11.02, n = 152/304 (50%); MHR < 11.02, n = 152/304 (50%).

**Table 4 cancers-17-02816-t004:** Multivariate Cox Proportional Hazards Model for Overall Mortality.

Variable	Hazard Ratio (95% CI)	*p*-Value
Signet-ring cell *present vs. absent*	1.77 (0.95–3.30)	0.072
Clinical Stage *II vs. I*	1.76 (0.34–9.06)	0.499
Clinical Stage *III vs. I*	1.61 (0.31–8.26)	0.572
Clinical Stage *IV vs. I*	1.65 (0.19–14.40)	0.648
Pathological Stage *II vs. I*	0.09 (0.01–0.68)	0.051
Pathological Stage *III vs. I*	0.36 (0.06–1.94)	0.236
Retrieved *LNs ≥ 15 vs. <15*	1.33 (0.62–2.85)	0.455
LVI *present vs. absent*	0.70 (0.24–2.06)	0.525
PNI *present vs. absent*	2.16 (0.73–6.38)	0.164
Monocyte-to-HDL *≥ 11.02 vs. <11.02*	1.04 (0.57–1.91)	0.876

−2 Log-likelihood = 383.01; global *p* = 0.023. HR, hazard ratio; CI, confidence interval; LNs, lymph nodes; LVI, lymphovascular invasion; PNI, perineural invasion.

**Table 5 cancers-17-02816-t005:** Multivariate Cox Proportional Hazards Model for Disease Progression.

Variable	Hazard Ratio (95% CI)	*p*-Value
Signet-ring cell *present vs. absent*	1.44 (0.86–2.42)	0.165
Clinical Stage *II vs. I*	0.62 (0.18–2.08)	0.439
Clinical Stage *III vs. I*	1.05 (0.33–3.31)	0.930
Clinical Stage *IV vs. I*	1.32 (0.21–8.68)	0.766
Neoadjuvant chemotherapy—*Yes vs. No*	1.47 (0.84–2.55)	0.169
Pathological Stage *II vs. I*	1.50 (0.35–6.37)	0.581
Pathological Stage *III vs. I*	3.04 (0.79–11.75)	0.106
LVI *present vs. absent*	1.40 (0.64–3.04)	0.387
PNI *present vs. absent*	1.64 (0.74–3.61)	0.217
Adjuvant chemotherapy—*Yes vs. No*	1.37 (0.76–2.48)	0.288
Monocyte-to-HDL *≥ 11.02 vs. <11.02*	1.93 (1.17–3.18)	0.010

−2 Log-likelihood = 651.47; global *p* < 0.001. HR, hazard ratio; CI, confidence interval.

## Data Availability

De-identified data underlying the findings can be made available from the corresponding author upon reasonable request and appropriate approvals.

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
