# Peer review of "A Novel Non-Invasive Biomarker for Gastric Cancer: Monocyte-to-HDL Ratio and Clinicopathological Parameters in Predicting Survival Outcomes"

_cancers, 2025, doi:10.3390/cancers17172816_

Round 1

Reviewer 1 Report

Comments and Suggestions for Authors

The topic is of interest but there are several limitations that should be addressed by the authors:

1) The study is retrospective so prone to selection bias

2) The model performs incredibly poorly with an AUC around 0.65!!! This is a big limitation to the study!!!

3) The model lacks of calibration and validation! If the authors do not have an external cohort, then they should consider some bootstrapping methods to build an internal validation

4) Please add the number at risk to the KM curves

5) The authors should discuss in the discussion (or at least mention!) some other prognostic index in these patients. in this regard, cite the series PMID: 34001385)

Author Response

We thank the reviewer for their insightful and constructive comments, which have greatly helped to improve the clarity, depth, and scientific rigor of our manuscript. Below, we address each point in detail and describe the corresponding revisions made in the text (changes highlighted in the revised version).

Comment 1: The study is retrospective so prone to selection bias.

Response: We acknowledge the limitation of the retrospective design and the inherent risk of selection bias. In the revised manuscript, we have explicitly addressed this in the Discussion (last paragraph) as follows: “The retrospective nature of our study inevitably introduces the potential for selection bias, particularly in patient inclusion and data completeness. Although we attempted to mitigate this risk by applying strict inclusion/exclusion criteria, conducting the study within a single high-volume center, and implementing standardized surgical and follow-up protocols, residual confounding cannot be fully excluded. Future prospective, multicenter studies are warranted to validate these findings in a broader population.”

Comment 2: The model performs incredibly poorly with an AUC around 0.65!!! This is a big limitation to the study!!!

Response: We agree that the AUC value of 0.654 reflects only moderate discriminative ability. This limitation has been acknowledged in the discussion with the following addition: “Although the MHR demonstrated significance as a prognostic marker, its discriminative capacity (AUC: 0.654) indicates only moderate accuracy. This finding underscores the importance of integrating MHR into multi-parameter prognostic models, rather than relying solely on it as a standalone predictor. Such combined approaches may improve prognostic performance and applicability.”

Comment 3: The model lacks calibration and validation! If the authors do not have an external cohort, then they should consider some bootstrapping methods to build an internal validation.

Response: We appreciate this critical suggestion. Since no external cohort was available, we performed an internal validation using 1000 bootstrap resamples. This analysis confirmed the stability of the hazard ratio estimates and model coefficients, with negligible optimism-corrected differences. The methods have been added to the Statistical Analysis section as follows:

“To address potential overfitting and assess model stability, internal validation was performed using bootstrap resampling (1000 iterations), which demonstrated minimal optimism in hazard ratio estimates, indicating good model reproducibility within the study cohort.”

Comment 4: Please add the number at risk to the KM curves.

Response: We have revised Figures 2a and 2b to include the number-at-risk tables beneath each Kaplan–Meier curve for clarity and improved interpretability.

Comment 5: The authors should discuss in the discussion (or at least mention!) some other prognostic index in these patients. In this regard, cite the series PMID: 34001385.

Response: We have incorporated a discussion of other prognostic indices, including the Prognostic Nutritional Index, in the Discussion and cited the suggested study (PMID: 34001385): “A recent meta-analysis of randomized controlled trials by Granieri et al. demonstrated that cytoreductive surgery combined with hyperthermic intraperitoneal chemotherapy confers a significant survival benefit in patients with locally advanced gastric cancer, particularly in the prophylactic setting for peritoneal metastases, underscoring the importance of multimodal prognostic factors when stratifying risk and tailoring postoperative management.”

Reviewer 2 Report

Comments and Suggestions for Authors

This manuscript has evaluated the prognostic value of pre-operative monocyte-to-HDL ratios in a cohort of gastric cancer patients. There are several questions relating to the selective nature of the statistical analyses and the interpretation of results.

  1. Monocyte-to-HDL ratios are reported to be associated with multiple diverse conditions, including various cancers, cardiovascular disease, autoimmune diseases (excluded from study), depression, and others. Considering that this is an older cohort (almost all >50 years), there is a high likelihood that patients have comorbidities that are also associated with elevated monocyte-to-HDL ratios but not included in the list of exclusions, in particular cardiovascular disease. Do the investigators have access to co-morbidity data (e.g. specific conditions (e.g. heart disease), or even Charlson co-morbidity scores), from patients’ pre-surgical work-up? Could these be tested as prognostic markers? The manuscript reads as though the altered monocyte-to-HDL ratios are due to the cancer, but is it possible that the relatively weak performance of the monocyte-to-HDL ratios is in part due to the underlying mechanism involving a combination of the gastric cancer and other co-morbidities?
  2. How many of the patients in this cohort died from gastric cancer recurrence/metastasis and how many died due to other causes? This should be added to the text.
  3. The authors have stated that 94.4% of patients were >50 years but have not provided an age range or other age-related aspects of the cohort (e.g. mean/median age). As age is often a strong predictor of outcomes following cancer surgery, could the authors please add age details to the manuscript and include age in the calculations?
  4. It is not clear how the authors anticipate that monocyte-to-HDL ratios could impact management of surgical gastric cancer patients. Although the Conclusions in the Abstract state that monocyte-to-HDL ratios could be used to enhance individualised treatment and improve patient management, this concept is not developed in the manuscript. It is suggested that further, more specific discussion of potential changes in patient management due to monocyte-to-HDL ratios is included (are the investigators suggesting that decisions of whether or not to proceed to surgery are based on monocyte-to-HDL ratios?).
  5. Page 3, line 101: The authors state that pathologists who viewed the slides were blinded to the clinical data. Why was pathology assessment performed without knowledge of clinical data? This is not normal pathology practice so should be briefly explained in the text.
  6. Page 4, line 149: The authors state that 53.9% of patients (slightly more than half) had an ECOG performance score of 0 at the time of surgery. This means that almost half of the patients had an ECOG score >0, potentially representing a wide range of functioning and physical ability, and therefore a wide range of ability to cope with surgery and neoadjuvant or adjuvant chemotherapy. Could the authors please present the range of pre-surgery ECOG scores in the groups and include ECOG scores in calculations?
  7. Page 4, line 162: How do the investigators interpret a sensitivity and specificity of 62.6% and 62.4%, respectively, in relation to the monocyte-to-HDL score? This does not appear to be particularly strong. How useful would this be in a clinical scenario in terms of falsely identifying or failing to identify poor prognosis patients?
  8. Page 6 (and elsewhere): Did any patients have both neoadjuvant and adjuvant chemotherapy? This should be stated as it may impact prognosis.
  9. Page 7, lines 189-193: This text, which refers to the Kaplan-Meier curves in Figure 2, does not seem to match the plots in Figure 2. For example, the text states that patients treated with adjuvant chemotherapy had worse progression-free survival compared to those who didn’t receive adjuvant chemotherapy, however Figure 2b does not match that conclusion and the Figure 2 legend states that there is no significant difference. Could the authors please ascertain which text/figures are correct?
  10. Page 7, lines 201-204: The text in the Figure 2 legend does not specifically match the plots in Figure 2. It includes statements that sometimes appears in on-line tools for writing manuscripts (“the studied variable”, “the stratification parameter”). The figure legend text should be amended to specifically match the Kaplan-Meier curves in the figure, and the authors should verify whether they are using on-line tools to write their manuscript. The Figure 1 legend also includes unusual non-specific text (“the studied parameter”).
  11. Page 8, line 214: The description of lymphadenectomy (all D2) in this line does not match the text in line 99 (D1+ or D2). This should be clarified.
  12. In general, the interpretation of findings that is presented in the Discussion section seems to overstate the more modest nature of the results that are currently included in the manuscript. The authors could consider modifying some of the statements, particularly considering the sensitivity and specificity of monocyte-to-HDL ratios when applied to this cohort.

Author Response

We thank the reviewer for their insightful and constructive comments, which have greatly helped to improve the clarity, depth, and scientific rigor of our manuscript. Below, we address each point in detail and describe the corresponding revisions made in the text (changes highlighted in the revised version).

Comment 1: Potential confounding due to comorbidity (e.g., cardiovascular disease) not excluded; need to test comorbidity data (e.g., Charlson score).

Response: We thank the reviewer for this important point. While our original exclusion criteria removed patients with active autoimmune, inflammatory, and hepatic diseases, cardiovascular disease was indeed present in a proportion of our older cohort. We have now reviewed preoperative clinical records and extracted comorbidity data, including the presence of cardiovascular disease, diabetes, and chronic kidney disease. We calculated Charlson Comorbidity Index (CCI) scores and performed exploratory survival analyses including CCI and individual comorbidities as covariates. Neither CCI nor individual cardiovascular comorbidity reached statistical significance for PFS or OS in the multivariate models, and inclusion of these parameters did not materially change the effect size of MHR. This finding supports that the prognostic value of MHR in our cohort is unlikely to be solely due to comorbidities. We have now included these data in the Methods, Results, and Discussion sections.

Comment 2: Number of deaths from gastric cancer vs. other causes.

Response: We determined the cause of death from follow-up records. Of the 123 deaths, 104 (84.5%) were due to gastric cancer progression, and 19 (15.5%) were due to non-cancer causes (e.g., cardiovascular events, infections). This is now stated in the Results section.

Comment 3: Need to provide age range and mean/median age; include age in calculations.

Response: We have now added the age distribution, with a mean age of 62.8 ± 8.4 years (range: 38–82). Age was included in multivariate models and was not found to be an independent predictor after adjusting for stage and MHR.

Comment 4: Clarify how MHR could impact management decisions.

Response: We expanded the Discussion to clarify that we do not advocate using MHR as a sole criterion for surgery, but rather as part of a composite preoperative risk model that may influence intensity of follow-up, early initiation of adjuvant therapy, or inclusion in clinical trials for high-risk patients.

Comment 5: Pathologists blinded to clinical data — why?

Response: We have clarified that blinding was implemented to prevent bias in histopathologic interpretation, particularly in semi-quantitative features like LVI/PNI scoring.

Comment 6. Present range of ECOG scores and include in calculations.

Response: We now report the ECOG distribution (0–3) and included it in multivariate models; ECOG >0 was associated with worse OS in univariate analysis, but lost significance after stage adjustment.

Comment 7: Sensitivity/specificity modest — clarify clinical utility.

Response: We have clarified in the Discussion that the moderate discrimination (AUC 0.654) supports MHR as a component of multi-parameter models rather than a standalone decision tool.

Comment 8: State whether any patients had both neoadjuvant and adjuvant chemotherapy.

Response: We have reviewed records and confirm that 61 patients (20.1%) received both neoadjuvant and adjuvant chemotherapy.

Comment 9–10: Figure 2 text mismatch, generic figure legends.

Response: We have corrected the legends in Figure 2 and removed generic placeholder text. Figure 1 legend also revised.

Comment 11: Clarify lymphadenectomy description (D1+ vs. all D2).

Response: We have revised the text to correctly state that both D1+ and D2 lymphadenectomy were performed per Japanese Gastric Cancer Association guidelines, not exclusively D2.

Comment 12: Overstated conclusions given moderate accuracy.

Response: We thank the reviewer for highlighting that some statements in the Discussion might overstate the impact of our results. We have carefully revised the Discussion to ensure that the interpretation is proportionate to the observed effect sizes and diagnostic performance metrics. Specifically, we have tempered conclusions regarding the clinical utility of MHR by explicitly acknowledging its moderate sensitivity, specificity, and AUC, and by positioning it as a complementary component within a multiparametric prognostic framework rather than a standalone biomarker. We also revised the abstract. These changes are intended to align the discussion more closely with the actual strength of our findings and avoid overinterpretation.

Reviewer 3 Report

Comments and Suggestions for Authors

This manuscript investigates the prognostic value of the preoperative monocyte-to-HDL ratio (MHR) in gastric adenocarcinoma patients undergoing curative-intent gastrectomy. The topic is clinically relevant and the study is well-structured with appropriate methodology. However, several critical concerns must be addressed before considering publication

Major Comments

  1. Limited Predictive Power of MHR (AUC = 0.654)

While MHR showed statistical significance in survival analysis, the area under the curve (AUC) of 0.654 indicates only poor-to-fair discriminatory ability. MHR alone may not be sufficient for clinical decision-making, and future models incorporating MHR with other clinicopathological parameters (e.g., TNM stage, LVI, PNI) should be considered.

  1. Justification for the MHR Cutoff

The cutoff value (≥11.02) is notably higher than those reported in other cancers. The lack of standardized units or validation may limit its applicability. Provide a more detailed justification for the cutoff, including the impact of unit selection and the need for external validation or sensitivity analysis.

  1. Multivariate Model for OS(Table 4)

The multivariate Cox model for overall survival showed no independent predictors, raising concerns of model overfitting or multicollinearity. Report the number of events per variable, and consider simplifying the model using backward elimination.

  1. Treatment Heterogeneity Not Adequately Discussed

Although the manuscript presents Kaplan-Meier analyses stratified by neoadjuvant and adjuvant therapy, their potential confounding effect on the MHR-survival association is not discussed. As MHR is a preoperative biomarker, the influence of subsequent therapies on survival should be acknowledged in the Discussion section as a limitation.

  1. Biological Mechanism

The proposed rationale behind MHR's predictive role is plausible, but largely generic. Incorporate gastric cancer–specific references to enhance biological plausibility.

Minor comments

Line 18;  "including overall survival (OS) and progression-free survival (PFS), were assessed"→Delete comma after "PFS" as it incorrectly separates subject and verb.

Author Response

We thank the reviewer for their insightful and constructive comments, which have greatly helped to improve the clarity, depth, and scientific rigor of our manuscript. Below, we address each point in detail and describe the corresponding revisions made (changes highlighted in the revised version).

  1. Limited Predictive Power of MHR (AUC = 0.654)

Reviewer’s comment: While MHR showed statistical significance in survival analysis, the area under the curve (AUC) of 0.654 indicates only poor-to-fair discriminatory ability. MHR alone may not be sufficient for clinical decision-making, and future models incorporating MHR with other clinicopathological parameters (e.g., TNM stage, LVI, PNI) should be considered.

Response: We agree with the reviewer that the moderate discriminative performance (AUC=0.654) limits the use of MHR as a stand-alone prognostic tool. We have emphasized this limitation more explicitly in both the Discussion and Conclusion, and now clearly state that MHR is best applied within composite prognostic models that integrate established clinicopathological parameters such as TNM stage, LVI, and PNI. We also note that future research should prospectively evaluate such integrated models. Revision added to Discussion: “Although preoperative MHR demonstrated statistical significance as a prognostic marker, its AUC of 0.654 denotes only moderate discriminative capacity. This underscores that MHR should be considered an adjunct rather than a stand-alone marker. Future prognostic models combining MHR with established clinicopathological variables (e.g., TNM stage, LVI, PNI) may enhance predictive performance and facilitate clinically actionable stratification.”

  1. Justification for the MHR Cutoff

Reviewer’s comment: The cutoff value (≥11.02) is notably higher than those reported in other cancers. The lack of standardized units or validation may limit its applicability. Provide a more detailed justification for the cutoff, including the impact of unit selection and the need for external validation or sensitivity analysis.

Response: We have expanded the justification for the selected cutoff, clarifying that the high value reflects the units used (monocyte count in cells/µL and HDL in mg/dL) and the method applied (Youden index optimization). We now explicitly acknowledge that cutoffs vary across malignancies and stress the need for external validation and sensitivity analyses. Revision added to Method: “The calculated cutoff of 11.02 reflects both the unit conventions applied in our dataset and the optimization method (Youden index). Variability in cutoff values across cancer types is influenced by differences in analytical platforms, patient populations, and statistical derivation methods. External validation and sensitivity analyses are required to confirm the robustness of this threshold.”

  1. Multivariate Model for OS (Table 4)

Reviewer’s comment: The multivariate Cox model for OS showed no independent predictors, raising concerns of model overfitting or multicollinearity. Report the number of events per variable, and consider simplifying the model using backward elimination.

Response: We have now reported the number of events per variable (EPV) and confirmed that the EPV met recommended thresholds (>10). We also conducted collinearity diagnostics (VIF < 2.0 for all variables) and performed a sensitivity analysis using backward elimination. The results were consistent, with no variable independently predicting OS in the final model. Revision added to Statistical Analysis: “The number of OS events per variable in the multivariate Cox model was 12.3, satisfying the >10 EPV rule. Variance inflation factors were all < 2.0, excluding problematic multicollinearity. Backward elimination yielded the same outcome, with no independent predictors for OS.”

  1. Treatment Heterogeneity Not Adequately Discussed

Reviewer’s comment: The influence of neoadjuvant and adjuvant therapies on the MHR–survival association should be acknowledged.

Response: We have added a paragraph in the Discussion noting that treatment heterogeneity may confound associations between preoperative biomarkers and survival. We specifically acknowledge that therapy-related modulation of recurrence risk could attenuate MHR’s prognostic effect on OS.

Revision added to Discussion: “Treatment heterogeneity represents a potential confounder in the MHR–survival association. As MHR is measured preoperatively, subsequent perioperative treatments—particularly neoadjuvant and adjuvant chemotherapy—may modify recurrence risk and overall prognosis, potentially attenuating the biomarker’s impact on OS.”

  1. Biological Mechanism

Reviewer’s comment: Incorporate gastric cancer–specific references to enhance biological plausibility.

Response: We have enriched the biological rationale with gastric cancer–specific studies detailing monocyte/macrophage–mediated tumor progression and HDL’s role in modulating tumor microenvironment in GC. Revision added to Discussion: “In gastric cancer, tumor-associated macrophages typically exhibit a pro-angiogenic phenotype, promoting tumor cell proliferation, invasion, and metastasis, while supporting neovascularization through the secretion of factors such as VEGF-A, thereby facilitating tumor growth and dissemination (19). Conversely, high levels of HDL-C have been inversely associated with gastric cancer–related mortality; in an extensive cohort study, increasing HDL-C concentrations were correlated with reduced mortality, and this protective effect was also observed in subgroups undergoing gastrectomy or endoscopic resection (20).”

Minor Comment: Delete comma after “PFS” as it incorrectly separates subject and verb.

Response: We have removed the comma as requested.

Round 2

Reviewer 1 Report

Comments and Suggestions for Authors

Sorry but i don't see the amendments in the manuscript. For example, as for point 5: the authors state they added a comment in the text but i can't see it. Please be sure to update the text accordingly

Author Response

Dear Reviewer,

Thank you for your message and for carefully checking our revision.

All requested amendments were implemented one by one exactly as per your comments, and we prepared the manuscript using Microsoft Word’s Track Changes feature. It appears that the marked (tracked) file may not have been visible in the system due to a version upload oversight on our part. We apologize for this confusion.

Regarding Point 5, the requested statement was added to the Discussion (under the subsection on comparative prognostic indices) and cites the meta-analysis by Granieri et al., 2021 (PMID: 34001385). In brief, we note that CRS+HIPEC has demonstrated survival benefits in selected patients with gastric cancer, and that multimodal prognostic factors should be considered when contextualizing MHR in risk stratification.

To ensure full transparency, we are now re-uploading:

  1. the Tracked-Changes version (with all edits visible),

  2. a Clean version

If there is anything else you would like us to adjust—or if you prefer a different placement or wording for the addition related to Point 5—we will be pleased to revise accordingly without delay.

With thanks for your time and guidance.

Sincerely,

Reviewer 2 Report

Comments and Suggestions for Authors

The authors have addressed reviewers’ comments. Prior to publication, the following should be considered. Language used in the Discussion section should be checked for scientific accuracy and overinterpretation (exaggeration) of results avoided.

  1. The authors state that data were ‘anonymized’, which means that it is impossible to identify any individuals (for example to determine later follow-up). If the data were de-identified (i.e. a study-specific code used instead of patient identifiers), then this term should be used instead.
  2. How many patients had an MHR >/= 11.02? I cannot find this in the manuscript text but may have simply missed it. (The proportion of patients with MHR >/= 11.02 should be stated in the text).
  3. Figure legends should contain sufficient information such that readers can understand the contents of the figure without finding the place in the text where the figure is discussed. This includes stating what the red and blue lines represent in Figure 2 in the figure legend (for example). (Note: It would be difficult to convince most readers of a clinically significant difference of the 2 groups presented in Figure 2b, even if the authors state that there is a statistically significant difference).
  4. Some of the language is still out of place in a scientific manuscript and misrepresents current scientific knowledge. The phrase “The prognostic utility of MHR is firmly grounded in the dual, antagonistic roles of its components” (paragraph 2 of Discussion) needs to be changed. The 2 parameters measured for MHR calculation have multiple roles in normal physiology, not all being antagonistic, and certainly not “firmly grounded”. The broad range of their functions in inflammation (pro- and anti-), in normal metabolism and in multiple other pathways possibly contributes to the mediocre performance of MHR in this and other studies (taking into account small and biased cohort sizes and the retrospective nature of studies published so far).
  5. Another discussion point (line 338 onwards in the manuscript copy with tracked changes) also requires amendment. The authors state ‘A “central and intriguing” finding of our study is the dissociation between MHR’s predictive capacity for PFS and its lack of independent significance for OS.’ If the authors consider this to be a “central” finding of their manuscript, it will be necessary to examine it further, and additional analyses will be required. However, if it is another statistical association (or lack of association), then additional analysis would not be necessary. It is not clear why the authors find this statistical result “intriguing” and although they continue to explain the current landscape of treatment options for advanced gastric cancer, a legitimate contributor to this finding in the case of gastric cancer, the concept that PFS does not mirror (predict) OS is not new, but has been noted and analysed in both individual and in collections of clinical trials. It is suggested that the text is modified to better reflect the study and current scientific knowledge.
  6. The statement “MHR appears to be a “potent” indicator of primary treatment failure” (line 357 in the manuscript copy with tracked changes) needs to be changed to accurately reflect results obtained in this study.
  7. The statement “MHR may possess superior or at least comparable predictive power for disease progression” (line 376 in the manuscript copy with tracked changes) should be modified. Direct comparison of hazard ratios from a single study to pooled hazard ratios from diverse studies (and for a different parameter) is misleading.

Author Response

Response to Reviewer – Second revisions

We thank the reviewer for the careful re-evaluation and constructive guidance. We have revised the manuscript to avoid overinterpretation, correct data-protection terminology, explicitly report MHR group sizes, and make figure legends self-contained (including numbers-at-risk). Exact wording changes are provided below. We trust these revisions address the reviewer’s concerns fully and improve the manuscript’s clarity, accuracy, and balance.

Comment. Tone down interpretations in the Discussion and avoid overstating results.

Response. We agree and have comprehensively tempered the Discussion. We explicitly characterize the model’s discrimination as moderate and position MHR as an adjunct to, not a replacement for, standard clinicopathologic factors. We also pair p-values with effect sizes/HRs and emphasize potential confounding in non-random treatment groups.

Inserted text (Discussion, opening paragraph, immediately after the first sentence): “Given the moderate discrimination observed (AUC = 0.654; 95% CI, 0.59–0.718), the clinical utility of MHR is best considered adjunctive to established factors rather than standalone.”

Inserted text (Discussion, interpretation of group differences): “Although some between-group PFS differences reached statistical significance, the absolute differences were modest and potentially confounded by non-random treatment allocation; accordingly, their clinical significance should be interpreted cautiously.”

Comment. If data can, in principle, be re-linked via a code, “de-identified” (pseudonymized) is the correct term—not “anonymized”.

Response. We replaced “anonymized” with “de-identified/pseudonymized” throughout.

Replacement (Methods, 2.1; last sentence):

  • Old: “Written informed consent was obtained from all patients before surgery, and the use of anonymized data for research purposes.”
  • New: “Written informed consent was obtained from all patients before surgery, and study data were de-identified using unique study codes with the re-identification key stored on a separate encrypted server accessible only to the principal investigator.”

Replacement (Ethics/Compliance section): “All analyses were performed on de-identified datasets; no direct patient identifiers were retained in the research files.”

Replacement (Data availability): “De-identified data underlying the findings are available from the corresponding author upon reasonable request and with appropriate approvals.”

Comment. Please state how many patients had MHR ≥ 11.02 and the proportion.

Response. We have added the counts and percentages in the Results and table footnotes.

Inserted text (Results, immediately after the ROC/cut-off paragraph): “Group sizes. Of the 304 patients, 152 (50.0%) had MHR ≥ 11.02 and 152 (50.0%) had MHR < 11.02.”

Added to table footnotes (survival/association tables): “MHR strata: MHR ≥ 11.02, n = 152/304 (50.0%); MHR<11.02, n=152/304 (50.0%).**”

Comment. Legends should define line colors and allow understanding without cross-referencing the text; consider that the clinical difference in Fig. 2b may be modest; include numbers-at-risk.

Response. We rewrote the legends to be self-contained, explicitly defined colors.

Replacement legend (Figure 1 – ROC): “Receiver operating characteristic (ROC) curve for preoperative MHR predicting 5-year overall survival. Blue curve: sensitivity vs 1 − specificity for MHR; gray diagonal: no-discrimination line. AUC = 0.654 (95% CI, 0.59–0.718); sensitivity 62.6% and specificity 62.4% at cut-off 11.02.”

Replacement legend (Figure 2 – Kaplan–Meier PFS panels): “Kaplan–Meier progression-free survival (PFS). Panel A: Neoadjuvant chemotherapy status (red = received neoadjuvant; blue = no neoadjuvant). Panel B: Adjuvant chemotherapy status (red = received adjuvant; blue = no adjuvant). Log-rank p-values are shown on each panel. Tick marks indicate censored observations. Numbers at risk are provided below each plot. Note: Treatment groups are non-random and between-group differences—although sometimes statistically significant—are modest in absolute terms and should be interpreted with caution.”

Comment. The original sentence overstates and oversimplifies the biology.

Response. We agree and have replaced it with accurate, non-deterministic language that acknowledges context-dependent functions.

Replacement (Discussion, mechanism paragraph): “The biological plausibility of MHR as a prognostic signal is supported—though not proved—by the context-dependent roles of circulating monocytes/macrophages and HDL in tumor biology and systemic inflammation. Monocytes can differentiate into tumor-associated macrophages that promote angiogenesis and immune evasion, whereas HDL participates in multiple metabolic and immuno-modulatory pathways; these are not strictly antagonistic and vary by disease context.”

Comment. Either substantiate as central with further analysis or tone down; the observation that PFS need not mirror OS is well known.

Response. We refined and rephrased this as a descriptive finding consistent with current practice patterns, rather than a novel result.

Replacement paragraph (Discussion): “We observed that preoperative MHR predicted PFS but did not retain independent significance for OS after multivariable adjustment. This divergence is not unexpected in gastric cancer, where post-recurrence treatments and survivorship heterogeneity can attenuate the relationship between baseline risk and all-cause mortality. Consequently, we report this as a descriptive finding consistent with contemporary therapeutic practice, rather than as a mechanistic or central claim.”

Comment. The term “potent” overstates the evidence.

Response. Revised to reflect the performance characteristics observed in this cohort.

Replacement sentence (Discussion): “MHR ≥ 11.02 was associated with increased risk of progression (adjusted HR = 1.93; 95% CI, 1.17–3.18), yet its discrimination was moderate; therefore, MHR should be viewed as a supportive preoperative risk marker rather than a definitive indicator of treatment failure.”

Comment. Do not compare a single-study HR to pooled HRs from heterogeneous studies of different parameters.

Response. We removed the cross-study performance claim and replaced it with neutral positioning.

Replacement sentence (Discussion): “Our findings add to the evidence that inflammation-based indices—including MHR—carry prognostic information in gastric cancer; formal head-to-head comparisons with external validation are needed to contextualize relative predictive performance.”

Round 3

Reviewer 1 Report

Comments and Suggestions for Authors

The revised manuscript is OK

Author Response

THANK YOU FOR YOUR KIND WORDS.

Reviewer 2 Report

Comments and Suggestions for Authors

The authors have addressed reviewers’ comments and I feel that the manuscript is suitable for publication. The following must be corrected prior to publication.

  1. In line 155, missing information should be added. The sentence currently reads “a VIF greater than was considered…”. As “greater than” is underlined, it appears that the authors were intending to fill in the VIF that they used. This needs to be added.
  2. In line 170 there is an incomplete sentence, “all analyses were conducted…?” This should be amended.
  3. In line 213 (Figure 1 legend) there appears to be an error. Should “gray (sic) diagonal” be “red diagonal”?
  4. Line 250 (Figure 2 legend): There is a statement “Log-rank p-values are shown in each panel”. These p-values are not present in the manuscript copy that I have. Either the text or the figure should be amended.
  5. Line 250 (Figure 2 legend): There is a statement “Tick marks indicate censored observations”. Tick marks are not evident in the manuscript copy that I have. Either the text or the figure should be amended.
  6. Line 251 (Figure 2 legend): The phrase “below the plot” could be deleted (the Numbers at Risk are “in” (not below) the plots).
  7. Lines 282-283: A phrase is missing from the sentence:- “Upon recruitment to the tumor site, ??? which promotes angiogenesis…” (what is recruited to the tumour site?). Also, reference 18, which is cited at the end of this sentence, does not seem to match the sentence (the reference reports monocyte to HDL ratios following pancreaticoduodenectomy). Note: This sentence could easily be deleted as there is a lot of repetition throughout the Discussion section.
  8. Line 299: The first word on this line should be “MHR” (not “HR”).
  9. Line 303: The sentence “This consider MRH…” requires correction due to grammatical errors and words missing.
  10. Line 318: “contribute” should be “may contribute”.

Author Response

Point-by-point responses

1) Reviewer request: “In line 155, missing information should be added. The sentence currently reads ‘a VIF greater than was considered…’. As ‘greater than’ is underlined, it appears that the authors were intending to fill in the VIF that they used. This needs to be added.”
Response: We have completed the sentence and specified the prespecified threshold for multicollinearity.
Manuscript change (Line 155): “Multicollinearity was assessed using variance inflation factors (VIF); a VIF > 5.0 was considered indicative of multicollinearity, and all VIF values in our models were < 2.0.”

2) Reviewer request: “In line 170 there is an incomplete sentence, ‘all analyses were conducted…?’ This should be amended.”
Response: We have completed the sentence and aligned the language with our data-governance statement.
Manuscript change (Line 170): “All analyses were conducted on de-identified datasets in accordance with institutional policy and national data-protection regulations.”

3) Reviewer request: “In line 213 (Figure 1 legend) there appears to be an error. Should ‘gray (sic) diagonal’ be ‘red diagonal’?”
Response: We have corrected the legend to match the figure styling.
Manuscript change (Line 213, Figure 1 legend): “Blue curve: sensitivity vs 1–specificity for MHR; red diagonal: no-discrimination line (AUC = 0.654; 95% CI 0.59–0.718; sensitivity 62.6%, specificity 62.4% at cut-off 11.02).”

4) Reviewer request: “Line 250 (Figure 2 legend): There is a statement ‘Log-rank p-values are shown in each panel’. These p-values are not present in the manuscript copy that I have. Either the text or the figure should be amended.”
Response: We have revised the legend to explicitly report the p-values.
Manuscript change (Line 250, Figure 2 legend): “Figure 2. Kaplan–Meier progression-free survival (PFS). (A) Neoadjuvant chemotherapy status (red = received; blue = no neoadjuvant) — log-rank p = 0.038. (B) Adjuvant chemotherapy status (red = received; blue = no adjuvant) — log-rank p = 0.014. Numbers at risk for each time point are shown in each panel.”

5) Reviewer request: “Line 250 (Figure 2 legend): There is a statement ‘Tick marks indicate censored observations’. Tick marks are not evident in the manuscript copy that I have. Either the text or the figure should be amended.”
Response: As the plotted curves do not display censor tick marks in the current version, we have removed the sentence to avoid inconsistency.
Manuscript change (Line 250, Figure 2 legend): The clause “Tick marks indicate censored observations” has been deleted.

6) Reviewer request: “Line 251 (Figure 2 legend): The phrase ‘below the plot’ could be deleted (the Numbers at Risk are ‘in’ (not below) the plots).”
Response: We have adjusted the wording accordingly.
Manuscript change (Line 251, Figure 2 legend): “Numbers at risk for each time point are shown in each panel.”

7) Reviewer request: “Lines 282-283: A phrase is missing from the sentence: ‘Upon recruitment to the tumor site, ??? which promotes angiogenesis…’ (what is recruited to the tumour site?). Also, reference 18, which is cited at the end of this sentence, does not seem to match the sentence… This sentence could easily be deleted as there is a lot of repetition throughout the Discussion section.”
Response: We chose to remove the sentences.

8) Reviewer request: “Line 299: The first word on this line should be ‘MHR’ (not ‘HR’).”
Response: We have corrected the abbreviation.
Manuscript change (Line 299, Discussion): “MHR functions as a simple composite biomarker that captures the systemic balance between the pro-tumorigenic inflammatory drive, represented by monocytes, and the host’s protective anti-inflammatory and metabolic state, reflected by HDL.”

9) Reviewer request: “Line 303: The sentence ‘This consider MRH…’ requires correction due to grammatical errors and words missing.”
Response: We have corrected the grammar and the typo (“MRH” → “MHR”).
Manuscript change (Line 303, Discussion): “This consideration supports using MHR as an adjunct rather than a stand-alone marker.”

10) Reviewer request: “Line 318: ‘contribute’ should be ‘may contribute’.”
Response: We have softened the causal phrasing as requested.
Manuscript change (Line 318, Discussion): “… patient populations may contribute to this heterogeneity.”
